# The Effects of a Home-Based Cardiac Rehabilitation Program via the Line Application on Functional Capacity and Quality of Life Among Open-Heart Surgery Patients: A Quasi-Experimental Study

**DOI:** 10.3390/healthcare13091051

**Published:** 2025-05-03

**Authors:** Suteetida Saensoda, Piyawan Pokpalagon, Suchira Chaiviboontham

**Affiliations:** 1Ramathibodi School of Nursing, Faculty of Medicine Ramathibodi Hospital, Mahidol University, Bangkok 10400, Thailand; suteetida.sae@student.mahidol.ac.th (S.S.); suchira.cha@mahidol.ac.th (S.C.); 2Department of Nursing, Rajavithi Hospital, Bangkok 10400, Thailand

**Keywords:** cardiovascular disease, cardiac rehabilitation, telerehabilitation, quality of life, self-care

## Abstract

**Background/Objectives**: This study aims to investigate the effects of a home-based cardiac rehabilitation (HBCR) program via a LINE application on functional capacity and quality of life (QOL) in open-heart surgery patients. **Methods**: This study involved 54 open-heart surgery patients divided into a control group and an experimental group (27 intervention, 27 control) using pair matching. Participants in the intervention group participated in the HBCR program, while the control group received standard care. Orem’s self-care theory was used as the conceptual framework. Functional capacity was measured via a six-minute walk test and the Duke Activity Status Index, while QOL was measured via the Thai version of the Short Form-36 Survey. **Results**: The findings from the study reveal that the patients who underwent open-heart surgery in the experimental group had significantly higher functional capacity compared to those in the control group (*p* < 0.05). Additionally, the overall QOL of the open-heart surgery patients in the experimental group was significantly better than that of the control group, alongside physical functioning, emotional roles, vitality, mental health, social functioning, and general health (*p* < 0.05). However, there were no significant differences between the two groups in terms of physical roles and bodily pain (*p* > 0.05). **Conclusions**: Using an HBCR program via the LINE application has the potential to enhance the at-home self-care ability of post-open-heart surgery patients, thus improving their functional capacity and QOL.

## 1. Introduction

Cardiovascular diseases (CVDs) are a major concern for global health and account for the highest mortality rate among non-communicable chronic illnesses and are the leading cause of death and disability worldwide, accounting for approximately 32% of total global deaths [1]. This trend continues to increase yearly, both globally and within many countries, including Thailand [2,3]. In Thailand, CVDs are the leading cause of death in the Thai population. By 2022, the mortality rate associated with CVD had increased from 31.82 in 2017 to 35.1 per 100,000 individuals [4]. Patients with CVD often require surgical treatment when pharmacological treatments are ineffective, symptoms worsen, or structural abnormalities in the heart and blood vessels require surgical correction [2,3].

Open-heart surgery is a common treatment for CVD [5], but it presents significant postoperative complications, continuing to impose significant burdens on patients and contributing to mortality, morbidity, and a diminished quality of life [6]. Cardiac rehabilitation (CR) is an evidence-based program comprising education, behavioral modifications, and exercise, designed to enhance outcomes in cardiovascular patients [7,8]. CR is crucial for reducing postoperative complications and improving outcomes, aiding recovery, and enhancing the overall quality of life for patients after surgery [9,10,11]. Nevertheless, a literature review has identified that postdischarge CR remains unsustainable due to multiple access barriers, including financial constraints, inadequate provider referrals, rural residency, lower socioeconomic status, and insufficient insurance coverage, with the primary challenge being the inconvenience of hospital-based rehabilitation programs [12,13,14].

The development of new delivery strategies is urgently required to enhance participation in cardiac rehabilitation, with home-based cardiac rehabilitation (HBCR) emerging as a potential approach to increasing patient engagement [7]. Growing evidence supports the implementation of home-based and technology-assisted CR as viable alternatives to conventional center-based programs, particularly in low and middle-income countries where access to cardiac rehabilitation services remains limited [8]. In contrast to center-based cardiac rehabilitation, which is administered within a medically supervised facility, HBCR utilizes remote coaching with indirect supervision of exercise, taking place primarily or entirely outside the conventional clinical setting [7]. Patients are expected to participate in self-directed rehabilitation activities at home while receiving professional guidance and continuous encouragement to sustain their CR practices [15]. Nurses play a crucial role in patient assessment and in planning and guiding home-based cardiac rehabilitation. Their support is crucial in helping patients with self-care and continuity in their rehabilitation, in turn leading to positive outcomes in terms of patient safety, feasibility, and program effectiveness. This empowerment enables patients to regain physical and emotional fitness, allowing them to function at their best [16,17]. These programs included activities such as self-monitoring and telehealth consultations aimed at enhancing accessibility, fostering continuous engagement in rehabilitation activities, gaining confidence in self-care, and reducing travel constraints and costs [18,19].

Information and communication technologies (ICTs) are considered essential tools for facilitating integrated primary care, particularly for patients with complex healthcare needs. According to the current literature, ICTs are pivotal in facilitating integrated primary care by supporting essential functions such as sharing information, promoting self-management, assisting in clinical decision-making, and enabling the delivery of healthcare services remotely [20]. HBCR programs have been developed by incorporating electronic activity notification systems and patient inquiry services related to CR to enable patients to participate actively in their CR [21]. A study on internet usage patterns in Thailand indicates that online social media platforms are widely employed across various age demographics, with the LINE application emerging as the predominant choice for conversations [22]. The LINE application’s benefits include the fact that it is an efficient long-distance communication tool that is user-friendly, convenient, and straightforward [23]. Additionally, it reduces expenses alongside the inconveniences faced by patients and their relatives related to travel [24]. Many patients in tertiary care hospitals who undergo open-heart surgery come from different provinces, presenting challenges in terms of travel costs. Therefore, using the LINE application for patient follow-up is an appropriate tool for continuous postcardiac surgery rehabilitation. Patients can access it easily and receive constant encouragement and support, promoting consistent cardiac rehabilitation. This concept aligns with the Theory of Self-Care (Orem’s), which asserts that individuals have an inherent ability to manage their health and provide self-care, provided they have access to the requisite knowledge, skills, and motivation. According to Orem, a self-care deficit occurs when individuals are unable to fulfill their self-care needs due to physical, psychological, or developmental limitations. In such instances, healthcare professionals intervene to offer support. Orem defined three nursing systems, including wholly compensatory, partially compensatory, and supportive–educative systems. Nurses’ roles in the supportive educational system come into play when the patient is ready to learn something but cannot do it without help and guidance. In this regard, using a nursing system in the form of a supportive–educative nursing system, which involves providing guidance, education, support, and encouragement to patients, helps patients make informed decisions about their self-care activities to develop their ability to manage their postsurgery recovery effectively and maintain a normal and healthy life [25].

Assessing functional capacity after cardiac surgery is critical for predicting rehabilitation outcomes [26,27]. The assessment of functional capacity is commonly conducted using the Six-Minute Walk Test (6 MWT) [28]. Studies have shown that an increase in patients’ walking distance beyond 70 m reflects a significant improvement in their quality of life [29]. Postcardiac surgery rehabilitation assessments demonstrate the broader impact on quality of life beyond just physical health [10,30]. Therefore, appropriate and continuous post-open-heart surgery rehabilitation can enhance functional capacity and overall quality of life [31,32].

The researchers endeavor to investigate the impact of an HBCR program via the LINE application among open-heart surgery patients on patients’ functional capacity and quality of life. The program includes a predischarge and postdischarge assessment at 12 weeks. The LINE application is utilized as the communication medium to provide knowledge about postsurgical self-care, behavior modification for health risk reduction, and postsurgical CR exercises. Patients are also monitored for any issues during the recovery process and are encouraged to regain confidence in their ability to recover their cardiac function appropriately and continuously.

## 2. Materials and Methods

### 2.1. Study Design

A quasi-experimental design was used for this study, employing a post-test-only design with a comparison group. The aim was to investigate the effectiveness of an HBCR program delivered via the LINE application with respect to the functional capacity and quality of life of open-heart surgery patients. The study recruited fifty-four participants from a tertiary care hospital in Bangkok, Thailand, between February 2023 and September 2023.

### 2.2. Setting and Sample

The participants consisted of patients with coronary artery disease after open-heart surgery who were admitted to a tertiary care hospital in Bangkok, Thailand. The participants had to meet the following inclusion criteria: they had to be between 18 and 60 years old, have undergone open-heart surgery for the first time, be oriented to time, place, and person, able to communicate and understand Thai language, have preoperative cardiac function categorized as New York Heart Association Functional Class I–III, not be handicapped or bedridden before hospital admission, and have a smartphone with the LINE application. The exclusion criteria included having severe postoperative complications (such as acute kidney injury requiring dialysis, severe liver disease, stroke, arrhythmias, or angina), reoperation to correct complications or abnormalities, mobility restrictions, and inability to perform a six-minute walk test. Pair matching was employed to match the characteristics of individuals within each pair of participant groups in an effort to ensure similarity between pairs in regard to gender—the individuals in each pair were of the same gender and of a similar age—and age, with a difference of no more than five years.

### 2.3. Sample Size

The determination of the sample size for this study was conducted using the power analysis method with the G*Power program [33]. The significance level was set at 0.05 and the power at 0.80. The effect size was based on a similar research study [34]. With an effect size of 0.80, the sample size was calculated to be 46 participants. To account for potential dropouts or withdrawals, a 15% increase was applied, yielding a total sample size of 54 participants, with 27 assigned to the experimental group and 27 to the control group. Throughout the 12-week intervention program, there were no dropouts or losses to follow-up recorded.

### 2.4. Research Procedures

#### 2.4.1. Intervention

An HBCR program was developed by researchers for open-heart surgery patients using the LINE application. This program was based on Orem’s Theory of Self-Care [25], the CR guidelines from the Heart Association of Thailand under the Royal Patronage of H.M. the King [35], and relevant literature reviews. The HBCR program was validated by five experts, who determined that the Content Validity Index was 1.0. The program’s duration was 12 weeks and included 10 contacts with the participants. The data for the intervention group were collected as shown in Figure 1. The program implementation comprised several key elements: (1) the researcher developed a 24-page electronic self-care handbook and a 15–20 min video about the HBCR program. The electronic handbook included information such as the disease, risk factors, smoking cessation, healthy eating, daily activities, dietary guidelines, stress management, medication adherence, wound care, self-monitoring of vital signs, cardiac rehabilitation, and the benefits of cardiac rehabilitation. The video about the HBCR program included warm-up exercises, walking routines, and cool-down exercises. The video also covered abnormal signs that indicate the need to stop exercising, along with corrective measures. The HBCR program involved 30 min sessions at least 3 days a week throughout the 12-week program. (2) The researcher assessed participants’ self-care and CR knowledge before hospital discharge. The participants then joined the program through the LINE application using an official account and accessed the video and electronic self-care handbook. The researcher explained the contents, demonstrated the CR activities, and then had the participants perform a return demonstration to ensure that they understood and could perform them correctly. (3) The researcher facilitated the exchange of learning and practical skills following hospital discharge. On days 3 and 7 postdischarge, the researcher provided video call follow-ups to monitor the correctness of exercise routines and encouraged continuous physical activity. (4) In weeks 2, 3, and 4 postdischarge, the researcher performed weekly video call follow-ups every Sunday. The researcher would review the participants’ self-care activities, asking about problems and obstacles they encountered with respect to CR and taking care of themselves, as well as encouraging discussion and creating opportunities for patients to express their concerns, providing emotional support and engaging in collaborative problem-solving during the LINE video calls. (5) In weeks 6, 8, and 10 postdischarge, biweekly LINE application visits were conducted on Sundays to assess the progress of the at-home cardiac rehabilitation. The researcher provided continuous follow-ups and support for CR through the LINE application. During the weeks in which participants did not have visits scheduled, they were expected to carry out the cardiac rehabilitation activities at home independently. If there were any issues, questions, or physical abnormalities, the participants could contact the researcher directly through the LINE application at any time of the day.

#### 2.4.2. Standard Care

The control group participants received usual nursing care according to the hospital’s standards for caring for patients undergoing open-heart surgery. They attended follow-up assessments at the outpatient cardiac surgery department two weeks after hospital discharge. During this visit, they received individualized instructions on CR from physical therapists and interprofessional teams, with each session lasting approximately 30 to 60 min. After that, they received phone calls in weeks 6 and 10 to monitor their progress in the physical rehabilitation exercise program. The participants underwent a 6 min walk test assessment in weeks 2 and 12 after hospital discharge. The researcher collected the data for the control group, as shown in Figure 1.

### 2.5. Research Instruments

All instruments were used with written permission obtained via email from the original authors.

The Personal Information Questionnaire was designed by the researcher and included information about gender, age, marital status, educational level, body weight, height, health status, and surgical history.

Functional capacity was assessed through the Six-Minute Walk Test (6 MWT) and physical activity was assessed using the Duke Activity Status Index (DASI). The 6 MWT was conducted according to the American Thoracic Society guidelines [29]. Patients’ heart rate, blood pressure, and oxygen saturation were monitored both before and after the test. If signs or symptoms of significant distress occurred during the 6 MWT, patients were permitted to stop and rest. The distance walked during the test was recorded in meters. The Thai version of the Duke Activity Status Index (DASI) was used to assess physical activity [36]. This instrument was modified and translated into Thai by Numpijit et al. [37]. The DASI is a 12-item instrument with two response options: “able to perform” and “unable to perform”. The DASI includes activities related to personal care, mobility, household chores, sexual function, and recreational activities, representing essential aspects of physical function. Each item is assigned a weight based on its metabolic equivalent (MET), ranging from 1.75 to 8.00 METs. The questions were sequenced based on the increasing order of energy expenditure required for each activity. If patients indicated that they could perform a particular activity, they were assigned the corresponding MET value. Total energy expenditure (MET) was calculated by summing the MET values for all activities. The total scores ranged from 0 to 58.20, based on the weights, with higher scores indicating better functional capacity. Previous studies have confirmed that the DASI has good reliability and validity [37,38].

The Thai version of the Short Form-36 (SF-36) was used to assess quality of life. The SF-36 was initially developed by The Mental Outcome Study and translated into Thai by Leurmarnkul and Meetam [39]. This questionnaire comprises 36 questions that are categorized into eight domains: physical functioning, physical roles, bodily pain, and general health are categorized under the physical health component, whereas vitality, social functioning, emotional roles, and mental health are grouped under the mental health component. An additional unscaled single item requires respondents to evaluate health changes over the past year. The item scores for each domain are coded, summed, and converted into a scale ranging from 0 (worst possible health state) to 100 (best possible health state) [40]. The total score is generated from the summation of all eight domains, with higher total scores indicating a better overall quality of life. Previous studies have confirmed that the Thai version of the SF-36 has good reliability and validity [41,42]. Cronbach’s α coefficient for every aspect of QOL exceeded 0.70, and all inter-item correlations exceeded 0.40 [42]. In this study, Cronbach’s α coefficient was 0.80.

### 2.6. Ethical Considerations

The researchers conducted ethically sound research on human subjects, with approval from the Ethics Committee of the Faculty of Medicine, Ramathibodi Hospital, Mahidol University(COA.MURA.2022/510), obtained on 10 September 2022, and from Rajavithi Hospital (approval No. 010/2566) on 27 January 2023. All participants were informed of the aims and details of this study. They had the option to participate or decline, and their participation was voluntary. They also signed consent forms to confirm their willingness to take part.

### 2.7. Statistic Analysis

The data underwent cleaning and coding prior to being entered into the computer program, and a level of statistical significance of 0.05 was adopted. Descriptive statistics were first used to evaluate the sociodemographic data. Subsequently, statistical analysis was conducted on the data as follows: (1) a chi-square test was utilized to compare the characteristics of both groups, and (2) independent *t*-tests were used to compare the mean differences in functional capacity (6 MWT and physical activity) and quality of life between the experimental group receiving an HBCR program and the control group receiving standard nursing care.

## 3. Results

Table 1 shows the demographic and clinical characteristics of the participants. The study involved a total of 54 individuals, with 27 in the control group and 27 in the experimental group. In the control group, 55.6% were men, compared to 77.8% in the experimental group. Most participants were aged between 51 and 60, with an average age of 47.11 years (SD = 9.82) in the control group and 46.85 years (SD = 11.17) in the experimental group. Most participants had heart valve diseases, with 66.7% in the control group and 55.6% in the experimental group. Most participants, 63.0%, across the two groups had heart conditions classified as New York Heart Association (NYHA) Class II. Most of the participants, 66.7%, in the control group and 66.7% in the experimental group had an ejection fraction greater than 50.0%. Regarding the surgical procedures, 66.7% of the control group and 55.6% of the experimental group underwent either valvular repair or valve replacement surgery. The chi-square test showed no significant differences in sociodemographic characteristics and clinical characteristics of the participants between the two groups (*p*> 0.05).

### 3.1. Functional Capacity

The functional capacity of all participants was assessed using the 6 MWT distance, while physical activity was assessed using the DASI. The mean 6 MWT distance walked by the participants enrolled in the HBCR program was 377.30 (SD = 77.16), which was higher than that walked by patients receiving standard care, with such a value being 326.19 (SD = 47.45). Additionally, the experimental group enrolled in the HBCR program had a higher mean physical activity of 42.71 (SD = 10.23) compared to 30.89 (SD = 9.24) for those receiving standard care. When comparing the means of the 6 MWT distance and physical activity of the control and the experimental groups using an independent t-test, it was found that the 6 MWT distance was significantly higher in the experimental group than in the control group (t = 2.93, *p* = 0.005, effect size = 0.80) (95% CI 16.13, 86.09), and the experimental group had a significantly higher mean physical activity score than that of the control group (t = 4.46, *p* = < 0.001, effect size = 1.22) (95% CI 6.50, 17.15). The details are presented in Table 2.

### 3.2. Quality of Life Among Open-Heart Surgery Patients

The mean scores of each dimension of QOL and the overall QOL of the control and experimental groups were compared using an independent t-test. The results showed that the participants in the experimental group had significantly higher average scores for overall quality of life (t = 4.55, *p* < 0.001, effect size = 1.25) (95% CI 49.37, 127.36), physical functioning (t = 3.57, *p* = 0.001, effect size = 0.97) (95% CI 6.57, 23.43), emotional roles (t = 2.34, *p* = 0.023, effect size = 0.97) (95% CI 1.24, 16.04), vitality (t = 3.61, *p* = 0.001, effect size = 0.98) (95% CI 4.12, 14.40), mental health (t = 5.34, *p* < 0.001, effect size = 1.45) (95% CI 6.01, 13.25), social functioning (t = 3.27, *p* = 0.002, effect size 0.72) (95% CI 4.65, 19.42), and general health (t = 5.05, *p* < 0.001, effect size = 0.72) (95% CI 13.06, 30.27) than those in the control group. However, there were no significant differences in the mean scores for physical roles (t = 0.58, *p* = 0.562, effect size = 0.16) (95% CI −6.78, 12.34) and the bodily pain dimension (t = 1.96, *p* = 0.056, effect size = 0.48) (95% CI −23, 18.94) between the control and experimental groups, as shown in Table 3.

## 4. Discussion

In this study, functional capacity was assessed using 6 MWT distance and physical activity to evaluate cardiovascular system recovery after open-heart surgery. Improving functional capacity is a crucial clinical outcome for CR related to secondary prevention of CVD [43]. After open-heart surgery, the experimental group that participated in the HBCR program showed significantly higher average 6 MWT distance and physical activity scores compared to the control group receiving standard care. The HBCR program, provided by the researchers, involved guiding patients before discharge from the hospital and providing continuous follow-ups over the ensuing 12 weeks. The HBCR program involved 30 min CR sessions at least 3 days a week throughout a 12-week course that consisted of exercise training involving practicing sitting, standing, and walking to facilitate efficient recovery and a rapid return to a normal state, thereby improving functional capacity [7]. HBCR programs are practical and enhance patient adherence to physical activity, thereby leading to an improvement in functional capacity. Additionally, patient education contributes to increasing levels of physical activity [43]. Consistent with previous studies, the HBCR program was shown to help improve functional capacity [18,44,45].

Moreover, the HBCR program uses an electronic booklet and online videos accessed through the LINE application to help patients perform exercises after open-heart surgery. These tools aim to enhance exercise capacity and build confidence in exercising. Patients can access the online video instructions at any time and perform the exercises independently at home after being discharged from the hospital. The combination of video-assisted programs and mobile-based follow-ups has greatly improved the effectiveness of home-based programs. These innovations have been key to better monitoring participant adherence and improving the quality of exercise execution at home. A previous Cochrane systematic review found that HBCR was associated with slightly higher adherence than center-based CR [46]. This approach highlights the potential of technological interventions in optimizing the outcomes of home-based health programs [43,47].

The present study also showed that the participants enrolled in the HBCR program had higher average scores for QOL in the physical functioning, emotional roles, vitality, mental health, social functioning, and general health dimensions than those in the control group receiving standard care. However, the mean scores for the physical roles and bodily pain dimensions between the control and experimental groups were not significantly different. The HBCR program, implemented through self-care guidance, exercise promotion, and encouragement to maintain a continuous record of CR progress, instills confidence in the patients’ own ability to self-manage their cardiac recovery. However, during the transition phase, when postsurgery patients return home, they often experience anxiety and uncertainty about self-managing their CR [18]. Combining the promotion of continuous CR with the LINE application to guide post-open-heart surgery CR can help alleviate travel limitations and costs [18,19]. The two-way communication facilitated through the LINE application allows patients to inquire about their CR, such as asking about exercise postures or sending wound pictures to assess infection symptoms, at any time. This active engagement empowers patients to actively participate in their physical and cardiac recovery, enhancing their self-confidence [18]. Open-heart surgery patients who have undergone major surgery require continuous physical, physiological, and psychological cardiac recovery in the long term [18,19].

The program empowered the participants to effectively promote self-care and CR at home postdischarge from the hospital. The results of this study suggest that providing tailored knowledge, offering self-care guidance, promoting mutual learning and exchange, stimulating continuous cardiac rehabilitation, and collaboratively addressing problems can build confidence in the patients’ own ability to self-manage their cardiac recovery after discharge [32]. Consequently, when patients experience appropriate and continuous cardiac recovery, it improves physical functioning and the overall quality of life. This aligns with the findings of previous studies [31], which have shown that CR programs that address other cardiovascular risk factors and provide education and social support could improve clinical outcomes in cardiopulmonary fitness, psychological factors, and the quality of life of patients with heart disease. This revealed that CR programs that provide knowledge, offer guidance on lifestyle adjustments, and promote healthy behaviors lead to improvements in the physical, mental, and social dimensions of quality of life [31,32]. The greater emphasis on self-monitoring, self-management, and unsupervised exercise in HBCR programs, in comparison to center-based CR, may facilitate a smoother transition from active intervention to lifelong disease self-management. However, further investigation is required to confirm this assumption [7]. In addition, studies have found that psychological support is a critical component of the CR process. The results of this study demonstrated that postcardiac surgery rehabilitation was associated with a greater prevalence of psychological discomfort symptoms relative to the general population. The extent of psychological distress, depression, and hostility was found to be correlated with limited improvements in rehabilitation outcomes. These findings underscore the importance of incorporating psychological support into CR programs to address these psychological challenges and improve the overall effectiveness of the rehabilitation process. Psychological distress can significantly impact a patient’s ability to engage fully in physical rehabilitation, and thus, psychological care should be integrated into CR programs to achieve comprehensive recovery [48]. The results from this study showed that the mean scores for the physical roles and bodily pain dimensions were not significantly different between the control and experimental groups. This outcome, which aligns with previous systematic reviews, demonstrates that HBCR and center-based CR offer comparable benefits regarding clinical outcomes and health-related quality of life [7,46]. Regarding physical roles, undergoing open-heart surgery has a significant impact on a patient’s physical problems due to the large incision wound caused by cutting open the sternum and intraoperative procedures, with around six to eight weeks required for the sternum and chest muscles to heal and the patient to regain physical function [3]. In terms of the bodily pain dimension, when patients attend a CR program that includes efficient respiratory and lung exercises, effective coughing techniques, and exercise training for CR, pain symptoms might arise following exercise. Therefore, in future studies, researchers should develop suitable pain management programs for patients undergoing open-heart surgery. Effective pain management leads to reduced limitations related to physical problems.

### Limitations and Recommendations

The study’s focus on a single site may limit its ability to fully represent all open-heart surgery patients, making it less applicable to other populations and countries. Additionally, while the program lasts 12 weeks, this intervention period may not fully capture the long-term effects of the HBCR program. In medical institutions, follow-up periods typically extend beyond 12 weeks, suggesting the need for further research to evaluate the long-term outcomes of continuous HBCR programs over six or 12 months. The study also used a quasi-experimental post-test-only research design, and conducting a randomized controlled trial (RCT) in future research to better test the effectiveness of the HBCR program should be considered. Furthermore, future research should consider employing logistic regression analysis to control for the effects of potential confounding variables that may influence postoperative recovery outcomes. Incorporating such analytical approaches would enhance the rigor of the findings and provide a more accurate estimation of the intervention’s effectiveness. It is also crucial to incorporate appropriate pain management strategies and engage family members in postsurgery care to improve outcomes in subsequent studies. Nurses and healthcare professionals could leverage the HBCR program via the LINE application as a model to enhance patient care, promote physical recovery, and improve quality of life, thereby improving access to CR for post-heart surgery patients. The results of our study suggest that an HBCR program is not only effective but also feasible, making it a valuable option for healthcare providers looking to enhance patient care in CR. Given the ease of implementation, reduced costs, and potential for wider accessibility, this model could be particularly beneficial in resource-limited settings or in the case of patients who face barriers to attending traditional rehabilitation programs in healthcare facilities (e.g., distance, time constraints, or physical limitations). In addition, future studies should include a cost-effectiveness analysis to evaluate the financial feasibility and long-term sustainability of the HBCR program, ensuring that it can be broadly adopted, particularly in low-resource settings. To scale this model, we recommend ensuring that healthcare providers, such as physiotherapists and nurses, are well-trained in delivering home-based rehabilitation programs. The intervention can be adapted to different healthcare settings by considering local resources and patient needs. For broader adoption, integrating this intervention into existing healthcare frameworks and ensuring collaboration among healthcare professionals is essential. For example, primary care providers could be key players in overseeing the rehabilitation process and providing necessary follow-up care. While our study focused on cardiac patients, the model’s flexibility makes it applicable to other populations with chronic health conditions, such as individuals with diabetes, chronic obstructive pulmonary disease (COPD), or poststroke rehabilitation.

## 5. Conclusions

This study demonstrated that using an HBCR program through the LINE application can significantly improve the functional capacity and quality of life of post-open-heart surgery patients at home, essential for their recovery and long-term well-being. These findings highlight the significance of HBCR and the role of technology in making it more accessible. The program that was developed spans from predischarge to 12 weeks postdischarge to provide knowledge, guidance, practical training, motivation, and continuous support to promote CR and enhance self-care abilities. Therefore, the HBCR program via the LINE application developed in this study can be adapted and implemented as a nursing practice guideline to facilitate appropriate self-care, continuous cardiac recovery, and improved patient outcomes after open-heart surgery. Additionally, it can also be applied to other patient groups requiring postsurgical cardiac rehabilitation. Future research should explore the application of this program to diverse patient populations, such as individuals with coronary artery disease, heart failure, or other cardiovascular conditions, to assess its broader effectiveness. The program could also be adapted for use in different healthcare settings, including outpatient clinics or community health centers, to expand its reach and accessibility. Additionally, studies should evaluate the long-term sustainability and cost-effectiveness of the program to determine its value in diverse healthcare contexts. Investigating the potential for integrating similar models into routine clinical practice would further enhance recovery across various medical conditions, supporting the widespread adoption of home-based rehabilitation programs.

## Figures and Tables

**Figure 1 healthcare-13-01051-f001:**
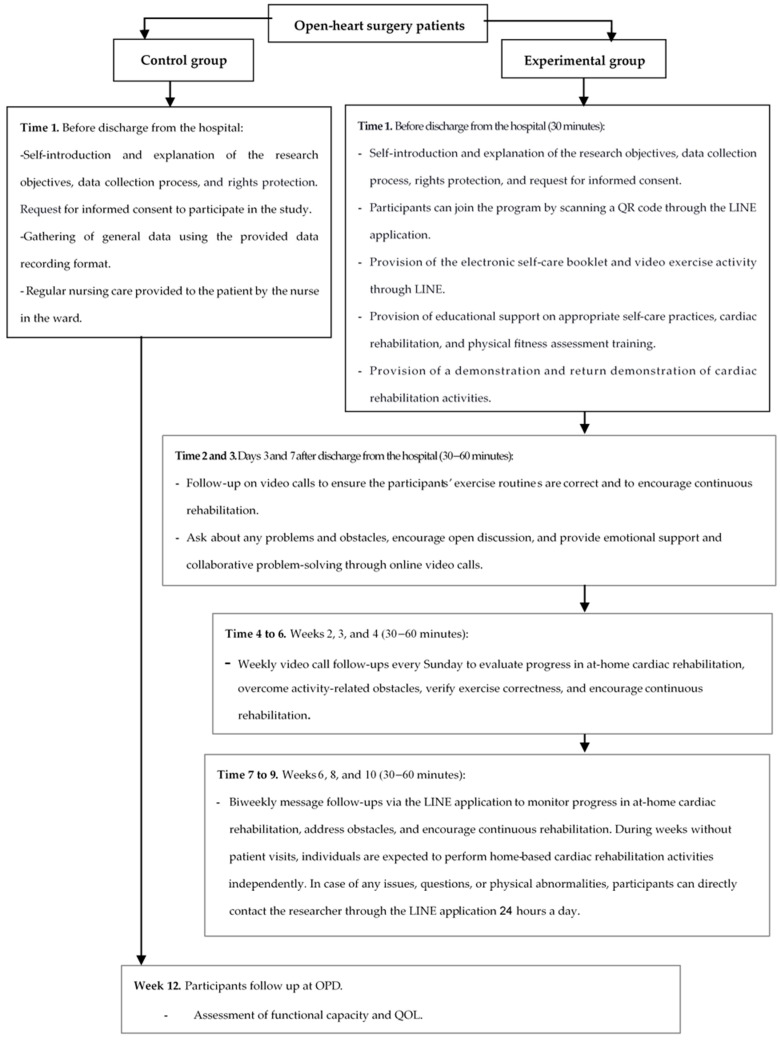
Research procedures and data collection in the control and experimental groups.

**Table 1 healthcare-13-01051-t001:** Demographic and clinical characteristics of the participants (N = 54).

	Experimental Group (n = 27), n (%)	Control Group (n = 27), n (%)	*p*-Value
**Gender**			
Male	21 (77.8)	15 (55.6)	0.803
Female	6 (22.2)	12 (44.4)	
**Age** (mean, SD) (Range 22–60, M = 46.98, SD = 10.74)	46.85 (11.2)	47.11 (9.8)	0.646
21–30	4 (14.8)	1 (3.7)	
31–40	4 (14.8)	6 (22.2)	
41–50	5 (18.5)	9 (33.3)	
51–60	14 (51.9)	11 (40.8)	
**Marital status**			0.424
Single	6 (22.2)	3 (11.1)	
Married	19 (70.4)	23 (85.2)	
Windowed	2 (7.4)	1 (3.7)	
**Education background**			0.503
Elementary education	4 (14.8)	5 (18.5)	
Secondary education	10 (37.0)	14 (51.9)	
Certificate level	5 (18.5)	2 (7.4)	
Bachelor’s degree or higher	8 (29.6)	6 (22.2)	
**Body Mass Index** (kg/m^2^) (mean, SD) (Range 15.61–38.79, M = 25.69, SD = 5.23)	25.62 (3.8)	25.77 (6.4)	0.436
**Occupation**			0.669
Unemployed	5 (18.5)	5 (18.5)	
Government officer	1 (3.7)	1 (3.7)	
Company employee	13 (48.2)	17 (63.0)	
Agriculture	2 (7.4)	2 (7.4)	
Merchant	5 (18.5)	1 (3.7)	
Other	1 (3.7)	1 (3.7)	
**Diagnosis**			0.432
Coronary disease	11 (40.7)	8 (29.6)	
Valve disease	15 (55.6)	18 (66.7)	
Vascular and valve disease	0	1 (3.7)	
DCM	1 (3.70)	0	
**Comorbidities** *			0.500
No underlying disease	7 (25.9)	8 (29.6)	
Presence of underlying disease	20 (74.1)	19 (70.4)	
Hypertension	15 (55.6)	17 (51.9)	
Diabetes	5 (18.5)	7 (25.9)	
Hyperlipidemia	14 (51.9)	13 (48.2)	
Chronic kidney disease	3 (11.1)	0	
Other	11 (40.7)	5 (18.5)	
**NYHA Class**			0.189
Class I	4 (14.8)	8 (29.6)	
Class II	17 (63.0)	17 (63.0)	
Class III	6 (22.2)	2 (7.4)	
**Ejection Fraction (EF)**			0.638
EF > 50%	18 (66.7)	18 (66.7)	
EF 40–50%	3 (11.1)	5 (18.5)	
EF < 40%	6 (22.2)	4 (14.8)	
**Surgical procedure**			0.198
OPCAB	8 (29.6)	2 (7.4)	
CABG	3 (11.1)	4 (14.8)	
Valve surgery	15 (55.6)	18 (66.7)	
Valve surgery + CABG	0	2 (7.4)	
Heart transplant	1 (3.7)	1 (3.7)	
**Cardiopulmonary bypass time** (hours) (mean, SD) (Range 0–5.00, M = 1.69, SD = 1.16)	1.55 (1.3)	1.84 (1.0)	0.517
**Surgery time** (hours) (mean, SD) (Range 2.15–9.45, M = 5.37, SD = 1.62)	5.41 (1.9)	5.33 (1.4)	0.500
**Length of stay** (days) (mean, SD) (Range 5–23, M = 10.63, SD = 3.32)	10.96 (3.2)	10.30 (3.5)	0.452

* One person has more than one disease. NYHA—New York Heart Association, EF—ejection fraction. OPCAB—off-pump coronary artery bypass grafting, CABG—coronary artery bypass grafting, DCM—dilated cardiomyopathy.

**Table 2 healthcare-13-01051-t002:** Comparison of functional capacity (6 MWT and physical activity) among open-heart surgery patients between the experimental group (n = 27) and the control group (n = 27).

Functional Capacity	Experimental Group (n = 27)		Control Group (n = 27)		Total (n = 54)		t	*p*-Value
	$\bar{X}$	SD	$\bar{X}$	SD	$\bar{X}$	SD		
6 MWT	377.30	77.16	326.19	47.45	351.74	68.49	2.93	0.005
Physical activity	42.71	10.23	30.89	9.24	36.80	11.35	4.46	0.000

Note: the significant level was set at *p* < 0.05.

**Table 3 healthcare-13-01051-t003:** Comparison of quality of life among open-heart surgery patients between the experimental group (n = 27) and the control group (n = 27).

Quality of Life	Experimental Group (n = 27)		Control Group (n = 27)		Total (n = 54)		t	*p*-Value
	$\bar{X}$	SD	$\bar{X}$	SD	$\bar{X}$	SD		
Physical functioning	86.48	13.07	71.48	17.48	78.99	17.06	3.57	0.001
Physical roles	93.52	16.40	90.74	18.54	92.13	17.40	0.58	0.562
Emotional roles	98.76	6.41	90.12	18.06	94.44	14.11	2.34	0.023
Vitality	75.37	9.08	66.11	9.74	70.74	10.43	3.61	0.001
Mental health	79.26	5.77	69.63	7.38	74.44	8.16	5.34	0.000
Social functioning	87.96	12.73	75.93	14.26	81.94	14.70	3.27	0.002
Bodily pain	81.57	19.82	72.22	14.94	76.90	18.01	1.96	0.056
General health	71.67	13.45	50.00	17.76	60.83	19.05	5.05	0.000
Overall QOL	674.60	65.86	586.23	76.54	630.42	83.61	4.55	0.000
Physical functioning	86.48	13.07	71.48	17.48	78.99	17.06	3.57	0.001
Physical roles	93.52	16.40	90.74	18.54	92.13	17.40	0.58	0.562
Emotional roles	98.76	6.41	90.12	18.06	94.44	14.11	2.34	0.023
Vitality	75.37	9.08	66.11	9.74	70.74	10.43	3.61	0.001
Mental health	79.26	5.77	69.63	7.38	74.44	8.16	5.34	0.000

## Data Availability

All original contributions from this study are contained within the article. For any further questions or information, please contact the corresponding author.

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
