# Peer review of "The Effects of a Home-Based Cardiac Rehabilitation Program via the Line Application on Functional Capacity and Quality of Life Among Open-Heart Surgery Patients: A Quasi-Experimental Study"

_healthcare, 2025, doi:10.3390/healthcare13091051_

Round 1
Reviewer 1 Report
Comments and Suggestions for Authors
Abstract of the Article
This manuscript investigates the effects of a home-based cardiac rehabilitation (HBCR) program delivered via the LINE application on functional capacity and quality of life in post-open-heart surgery patients. The study is guided by Orem’s self-care theory and uses a quasi-experimental with a post-test-only design. It includes 54 participants, evenly divided into control and experimental groups. The findings indicate that the HBCR program significantly improves both functional capacity and quality of life compared to standard care.
Strengths of the Paper
The manuscript addresses an important issue, particularly in the context of low- and middle-income countries where access to center-based cardiac rehabilitation remains limited. The use of LINE as a communication platform is a pragmatic choice, considering its popularity in Thailand.
The intervention is well structured and its application is clearly described. The authors provide a detailed description of the steps taken from pre-discharge to post-discharge follow-up. The study also relies on validated tools, such as the 6-Minute Walk Test, the Duke Activity Status Index, and the Thai version of the SF-36, to measure outcomes, which supports the robustness of the results. The statistical tests employed are appropriate, and the results are clearly presented.
The paper is easy to follow, and the language is clear. The figures and tables are useful and well integrated into the manuscript.
Points for Improvement
- Clarification on Dropouts
The sample size is described as 54 participants divided into two groups. However, there is no mention of dropouts or losses to follow-up during the 12-week intervention. If no participants dropped out, this should be explicitly stated. If there were dropouts, the authors should report the number and provide reasons.
- Development of the Theoretical Background
The manuscript mentions that Orem’s self-care theory serves as the conceptual framework. However, the theory is not developed in the manuscript, nor is it sufficiently connected to the design of the intervention. I don’t know this theory in particular so that I would be curious to understand its main components and how the authors developed the intervention based on it. I am convinced my situation is the case for most of readers.
- Methodological Limitations
The use of a post-test-only design is a limitation that deserves more attention. Without pre-intervention data, it is difficult to fully assess the impact of the intervention. Even though pair-matching was used to create equivalent groups, the lack of baseline measures for the main outcomes (6MWT, Physical activity, QOL) is problematic. The authors should reflect more on this limitation in the discussion.
- Reporting of Effect Sizes
While the authors report statistically significant differences, effect sizes are not provided. Including effect sizes and confidence intervals would allow readers to better understand the magnitude of the intervention’s impact and its practical significance.
- Implications for Practice and Future Research
The conclusion presents the main findings clearly but would be stronger with a more developed section on implications for practice. Given the promising results and the feasibility of the intervention, the authors should elaborate on how this model could be scaled or adapted in other settings. The recommendation to extend the intervention to other patient groups is interesting but remains vague. More concrete suggestions would be welcome.
Author Response
|
Reviewer 1 Comments 1: Clarification on Dropouts: The sample size is described as 54 participants divided into two groups. However, there is no mention of dropouts or losses to follow-up during the 12-week intervention. If no participants dropped out, this should be explicitly stated. If there were dropouts, the authors should report the number and provide reasons. |
|
Response 1: Thank you for pointing this out. We agree with this comment. Accordingly, a clarification has been added to explicitly indicate that no participant dropped out or were lost follow-up throughout the 12-week intervention period. This statement has been added to the Methods section on page 4, paragraph 3, line 156-157 in the revised manuscript.
Comment 2: Development of the Theoretical Background: The manuscript mentions that Orem’s self-care theory serves as the conceptual framework. However, the theory is not developed in the manuscript, nor is it sufficiently connected to the design of the intervention. I don’t know this theory in particular so that I would be curious to understand its main components and how the authors developed the intervention based on it. I am convinced my situation is the case for most of readers. Response 2: Agree. We have, accordingly, revised the manuscript to provide a more detailed explanation of Orem's Self-Care Theory and its main components. We have also clarified how this theory was used to inform the design of the intervention. These revisions are intended to help readers better understand the conceptual framework and its relevance to the intervention. The updated text can be found in the revised manuscript on page 3, lines 94-103. [Updated text in the manuscript if necessary: "In this study, we utilized Orem’s Self-Care Theory as the guiding framework for the intervention. Orem’s theory emphasizes the importance of individuals' ability to engage in self-care activities to maintain, restore, or improve their health. The main components of the theory include self-care, self-care deficit, and nursing systems. The intervention was designed to address identified self-care deficits in patients by providing necessary guidance and support, enabling them to enhance their self-care abilities, particularly during post-open heart surgery recovery."]
Comments 3: Methodological Limitations: The use of a post-test-only design is a limitation that deserves more attention. Without pre-intervention data, it is difficult to fully assess the impact of the intervention. Even though pair-matching was used to create equivalent groups, the lack of baseline measures for the main outcomes (6MWT, Physical activity, QOL) is problematic. The authors should reflect more on this limitation in the discussion. Response 3: We agree that the use of a post-test-only design is a limitation of this study. To mitigate this limitation, we employed pair-matching based on age and gender to create comparable groups. Additionally, we conducted statistical comparisons of sociodemographic and clinical characteristics between the two groups to ensure equivalence at baseline as much as possible. We have also addressed this issue in the "Limitations and Recommendations" section of the manuscript, where we discuss the constraints of using this research design. Additionally, we recommend conducting a randomized controlled trial (RCT) in future research to better assess the effectiveness of the home-based cardiac rehabilitation program.
Comments 4: Reporting of Effect Sizes: While the authors report statistically significant differences, effect sizes are not provided. Including effect sizes and confidence intervals would allow readers to better understand the magnitude of the intervention’s impact and its practical significance. Response 4: Thank you for your valuable comment regarding the reporting of effect sizes. We appreciate your suggestion to include effect sizes and confidence intervals to provide a better understanding of the magnitude and practical significance of the intervention. In response to your comment, we have calculated and included the appropriate effect sizes (Cohen's d) for the independent t-test comparisons. We also report 95% confidence intervals for the effect sizes to provide further clarity on the precision of these estimates. We have updated the manuscript to include Cohen's d values for all significant comparisons, accompanied by 95% confidence intervals. These revisions have been included in the manuscript and can be found on page 10 paragraph 1, lines 333-335 and paragraph 2, lines 347-355.
Comment 5: Implications for Practice and Future Research: The conclusion presents the main findings clearly but would be stronger with a more developed section on implications for practice. Given the promising results and the feasibility of the intervention, the authors should elaborate on how this model could be scaled or adapted in other settings. The recommendation to extend the intervention to other patient groups is interesting but remains vague. More concrete suggestions would be welcome. Response 5: Thank you for your insightful comment regarding the implications for practice and future research. In light of your suggestion, we have revised the conclusion to provide a more detailed discussion of how this intervention could be scaled and adapted for use in other settings. We have also included more specific recommendations for extending the intervention to other patient groups. These revisions have been included in the manuscript and can be found on page 13, paragraph 1, lines 457-480. “The results of our study suggest that a home-based cardiac rehabilitation program is not only effective but also feasible, making it a valuable option for healthcare providers looking to enhance patient care in cardiac rehabilitation. Given the ease of implementation, reduced costs, and potential for wider accessibility, this model could be particularly beneficial in resource-limited settings or for patients who face barriers to attending traditional rehabilitation programs in healthcare facilities (e.g., distance, time constraints, or physical limitations). To scale this model, we recommend ensuring that healthcare providers, such as physiotherapists and nurses, are well-trained in delivering home-based rehabilitation programs. The intervention can be adapted to different healthcare settings by considering local resources and patient needs. For broader adoption, integrating this intervention into existing healthcare frameworks and ensuring collaboration between healthcare professionals is essential. For example, primary care providers could be key players in overseeing the rehabilitation process and providing necessary follow-up care. While our study focused on cardiac patients, the model's flexibility makes it applicable to other populations with chronic health conditions, such as individuals with diabetes, chronic obstructive pulmonary disease (COPD), or post-stroke rehabilitation.” |

Reviewer 2 Report
Comments and Suggestions for Authors
Dear Authors,
I appreciate your paper, which reported a relevant topic.
I have some suggestions:
- The introduction is too long and detailed; part of it should be better detailed in the discussion
- Which is the meaning of "quasi" experimental design
- Why do you refer to the guidelines of Thailand (ref 35) instead of the international ones
- You need to detail the LINE application
- Line 183: what do you mean by "provide emotional support"? Was it a therapeutic intervention?
- Line 199, it is not clear if the control group also had an exercise plan
- In the discussion, I would suggest more details on the aspects related to psychological support and therapy, which are fundamental parts of rehabilitation. See and cite Ricci M, Pozzi G, Caraglia N, Chieffo DPR, Polese D, Galiuto L. Psychological Distress Affects Performance during Exercise-Based Cardiac Rehabilitation. Life (Basel). 2024 Feb 8;14(2):236. doi: 10.3390/life14020236. PMID: 38398745; PMCID: PMC10890595.
Comments on the Quality of English Language
English language should be improved
Author Response
Comments 1:
The introduction is too long and detailed; part of it should be better detailed in the discussion
Response 1: Thank you for your constructive feedback regarding the length and level of detail in the introduction. We appreciate your suggestion to streamline the introduction and shift some of the detailed information to the discussion section. In response to your comment, we have revised the introduction to make it more concise by removing some of the detailed background information that is more appropriate for the discussion.
Comment 2:
Which is the meaning of "quasi" experimental design
Response 2: Thank you for pointing this out. A "quasi-experimental design" refers to a type of research design that seeks to evaluate the causal impact of an intervention or treatment but does not use random assignment of participants to different groups, as in a true experimental design. Instead, quasi-experimental designs use existing groups or naturally occurring events, and the researcher observes the effects of an intervention on those groups.
In essence, while a quasi-experimental design still looks for cause-and-effect relationships, it lacks some of the controls of a true experimental design (such as random assignment), which makes it more vulnerable to certain biases or confounding factors.
Comment 3:
Why do you refer to the guidelines of Thailand (ref 35) instead of the international ones
Response 3: Thank you for pointing this out. We chose to reference the Thai national cardiac rehabilitation guidelines because our study was conducted in Thailand, and our intervention was designed to fit the local healthcare context, resources, and patient population. These national guidelines are based on international standards but are specifically tailored to the cultural and economic realities of delivering home-based cardiac rehabilitation in Thailand. Additionally, using local guidelines ensures that our approach is clinically relevant and feasible for real-world practice in the country.
Comment 4: You need to detail the LINE application
Response 4: Thank you for pointing this out. We have provided a detailed explanation of how the LINE application was used in research procedure of the study. Participants accessed the program through an official account, which included videos and a self-care handbook. The researcher provided follow-up through weekly and biweekly video calls to monitor exercise routines, offer support, and address any issues. These follow-ups were designed to ensure correct practice and encourage continuous engagement in rehabilitation activities.
Comment 5: Line 183: what do you mean by "provide emotional support"? Was it a therapeutic intervention?
Response 5: Thank you for pointing this out. According to the components of the cardiac rehabilitation program outlined by The Royal College of Physiatrists of Thailand, the program consists of three main elements: 1) provision of knowledge on post-surgical self-care practices, 2) cardiac fitness exercises, and 3) psychological support. Therefore, emotional support was incorporated into the intervention group as part of the psychological support component. This support aimed to address the emotional well-being of patients, providing encouragement and addressing concerns during their recovery process. It was not intended as a therapeutic intervention but rather as an integral part of the overall rehabilitation program.
Comment 6: Line 199, it is not clear if the control group also had an exercise plan
Response 6: Thank you for pointing this out. To clarify, the control group received usual nursing care according to the hospital’s standards, which included routine follow-up assessments, individualized cardiac rehabilitation instructions from physical therapists, and phone calls at weeks 6 and 10 to monitor progress. They also underwent a 6-minute walk test at weeks 2 and 12 after discharge. However, they did not receive continuous follow-up visits to assess and address any issues or obstacles related to their cardiac rehabilitation program.
Comment 7: In the discussion, I would suggest more details on the aspects related to psychological support and therapy, which are fundamental parts of rehabilitation. See and cite Ricci M, Pozzi G, Caraglia N, Chieffo DPR, Polese D, Galiuto L. Psychological Distress Affects Performance during Exercise-Based Cardiac Rehabilitation. Life (Basel). 2024 Feb 8;14(2):236. doi: 10.3390/life14020236. PMID: 38398745; PMCID: PMC10890595.
Response 7: Thank you for your valuable comment. We have revised the discussion as per your suggestion, and added more details on psychological support and therapy, which are indeed fundamental aspects of rehabilitation. The updated content can be found on page 12, lines 425-435. Additionally, we have cited the article you recommended by Ricci et al. (2024) to further support our discussion on psychological distress in cardiac rehabilitation.

Reviewer 3 Report
Comments and Suggestions for Authors
This manuscript by Suteetida Saensoda et.al investigate the impact of a home-based cardiac rehabilitation (HBCR) program delivered via the LINE application on the functional capacity and quality of life (QOL) of patients who have undergone open-heart surgery. However, some concerns need to be addressed.
1.The quasi-experimental design limits the strength of the conclusions. Therefore, the author should elaborate on the principles of grouping, whether age or gender matching is performed, and avoid artificial grouping bias
2. The sample size is relatively small, and the study is limited to a single tertiary care hospital in Bangkok, Thailand. This limits the generalizability of the findings to other populations and settings. Moreover, The study only employs a post-test-only comparison group, lacking control of confounding factors.
3. The study duration of 12 weeks is relatively short. Longer-term follow-up is needed to assess the sustained impact of the HBCR program on functional capacity and quality of life.
4. While the HBCR program is described in detail, including the use of the LINE application, electronic self-care handbook, and video materials, more information on the specific content and delivery of the intervention would be beneficial.
5.The study found no significant differences between the groups in terms of physical roles and bodily pain. This suggests that pain management may be an area requiring further attention in HBCR programs.
6. Cost-effectiveness Analysis**: The study does not address the cost-effectiveness of the HBCR program. Given the potential for technology-based interventions to reduce healthcare costs, a cost-effectiveness analysis would provide valuable insights into the feasibility and sustainability of implementing such programs on a larger scale.
Comments on the Quality of English Language
Some typos and minor grammar errors need to be corrected
Author Response
Comments 1:
The quasi-experimental design limits the strength of the conclusions. Therefore, the author should elaborate on the principles of grouping, whether age or gender matching is performed, and avoid artificial grouping bias
Response 1: Thank you for pointing this out. We agree with this comment. Accordingly, in this study, pair matching was employed to ensure similarity between participants in terms of gender and age. Specifically, participants in each pair were matched by gender and had an age difference of no more than five years. This approach was used to reduce potential bias and enhance the comparability of the groups. Although the quasi-experimental design has inherent limitations, the use of pair matching aimed to address some of the concerns related to grouping bias. We will provide further elaboration on this methodology in the revised manuscript to ensure greater clarity.
Comment 2:
The sample size is relatively small, and the study is limited to a single tertiary care hospital in Bangkok, Thailand. This limits the generalizability of the findings to other populations and settings. Moreover, The study only employs a post-test-only comparison group, lacking control of confounding factors.
Response 2: Thank you for pointing this out. We acknowledge that the sample size in this study is relatively small, and the research was conducted at a single tertiary care hospital in Bangkok, Thailand, which limits the generalizability of the findings. However, it is important to note that this study serves as a preliminary investigation into the feasibility and effectiveness of a home-based cardiac rehabilitation (HBCR) program, particularly using technology such as the LINE application. Given the context of our research, we believe that this study provides valuable insights into the potential for implementing HBCR in similar settings, and it serves as a foundation for future studies with larger, more diverse populations. To improve the generalizability of our findings, we recommend conducting future studies in multiple hospitals across different regions of Thailand, as well as in low- and middle-income countries where access to cardiac rehabilitation services is limited.
Regarding the use of a post-test-only design, we recognize this as a limitation of the study. This design was chosen due to the practical limitations of implementing a pre-test in a clinical setting, especially with patients recovering from surgery. A post-test-only design allowed us to assess the immediate outcomes of the HBCR program without placing additional burden on participants during their recovery period. However, we have addressed this limitation in the "Limitations and Recommendations" section of the manuscript, where we discuss the constraints of this research design. To improve upon this, we recommend conducting a randomized controlled trial (RCT) in future research to better assess the effectiveness of the home-based cardiac rehabilitation program and to control for confounding factors.
Comment 3: The study duration of 12 weeks is relatively short. Longer-term follow-up is needed to assess the sustained impact of the HBCR program on functional capacity and quality of life.
Response 3: Thank you for pointing this out. This study investigates the effects of home-based cardiac rehabilitation (HBCR) after open-heart surgery, focusing on the outpatient cardiac rehabilitation (Phase II) (from leaving the hospital to outpatient rehabilitation, typically lasts 3 to 6 weeks, though some programs may extend up to 12 weeks), which is the transitional phase. Post-surgery patients returning home often experience anxiety and uncertainty about self-managing their cardiac rehabilitation. Therefore, the study focused on this phase, with the duration starting from discharge and lasting approximately 8-12 weeks. However, we agree that longer-term outcome studies are necessary to assess the sustained impact of the HBCR program on functional capacity and quality of life. This has been added to the 'Recommendations' section of the manuscript.
Comment 4: While the HBCR program is described in detail, including the use of the LINE application, electronic self-care handbook, and video materials, more information on the specific content and delivery of the intervention would be beneficial.
Response 4: Thank you for your valuable comment. We agree that providing more details on the specific content and delivery of the intervention would be beneficial. We have clarified this in the "Research Procedure" section, page 4, paragraph 4, lines 159-195. This includes the use of the LINE application, the self-care handbook, and video materials as key components of the home-based cardiac rehabilitation program. These resources provide comprehensive information on disease management, risk factors, smoking cessation, healthy eating, daily activities, stress management, and more. The video materials also offer detailed instructions on cardiac rehabilitation exercises, enabling patients to follow them effectively at home. These interventions are designed to help patients manage their rehabilitation process efficiently during the program.
Comment 5: The study found no significant differences between the groups in terms of physical roles and bodily pain. This suggests that pain management may be an area requiring further attention in HBCR programs.
Response 5: Thank you for pointing this out. We agree with your suggestion. The study found no significant differences between the groups in terms of physical roles and bodily pain. Therefore, we have discussed this in the discussion section and recommend that future studies focus on developing effective pain management strategies for post-heart surgery patients to reduce physical limitations.
Comment 6: Cost-effectiveness Analysis**: The study does not address the cost-effectiveness of the HBCR program. Given the potential for technology-based interventions to reduce healthcare costs, a cost-effectiveness analysis would provide valuable insights into the feasibility and sustainability of implementing such programs on a larger scale.
Response 6: Thank you for your thoughtful comment regarding the inclusion of a cost-effectiveness analysis in our study. We agree that assessing the cost-effectiveness of the home-based cardiac rehabilitation (HBCR) program is an important aspect, especially considering the potential for technology-based interventions to reduce healthcare costs and improve accessibility. Due to the scope of the current study and the resources available, we were unable to conduct a comprehensive cost-effective analysis. The focus of this pilot study was to assess the clinical outcomes and feasibility of implementing a technology-assisted intervention in a specific patient population. To address this gap, we propose that future studies should include a cost-effectiveness analysis.

Round 2
Reviewer 3 Report
Comments and Suggestions for Authors
The author has responded to each review comment individually, but has only offered explanations without making substantial modifications. In my opinion, these modifications are far from sufficient. For example, numerous confounding factors can influence the improvement of patients' postoperative conditions. Therefore, it is essential to select appropriate research methods, such as logistic regression analysis, to adjust for the impact of these confounding factors on the results. Additionally, I fully understand that intervention procedures under the supervision of medical personnel cannot be prolonged indefinitely (e.g., 12 weeks). However, their long-term effects still need to be evaluated. In the current follow-up process in medical institutions, the duration of patient follow-up is not limited to just 12 weeks.
Author Response
|
Comments 1: The author has responded to each review comment individually, but has only offered explanations without making substantial modifications. In my opinion, these modifications are far from sufficient. For example, numerous confounding factors can influence the improvement of patients' postoperative conditions. Therefore, it is essential to select appropriate research methods, such as logistic regression analysis, to adjust for the impact of these confounding factors on the results. Additionally, I fully understand that intervention procedures under the supervision of medical personnel cannot be prolonged indefinitely (e.g., 12 weeks). However, their long-term effects still need to be evaluated. In the current follow-up process in medical institutions, the duration of patient follow-up is not limited to just 12 weeks. |
|
Response 1: Thank you for your valuable comment. We acknowledge the importance of addressing confounding factors in our study. In response to your suggestion, we have revised our discussion and highlighted the potential use of logistic regression analysis in future research to control for the effects of confounding variables that could influence postoperative recovery outcomes. We believe that incorporating such analytical methods would significantly enhance the rigor of the study and lead to a more accurate estimation of the intervention's effectiveness. The updated content can be found on page 13, paragraph 1, line 459-463. Regarding your comment about the intervention period, we fully agree that a 12-week follow-up may not capture the long-term effects of the Home-Based Cardiac Rehabilitation (HBCR) program. While the current follow-up period was limited to 12 weeks, we have acknowledged that in medical practice, follow-up typically extends beyond 12 weeks. In response, we recommend that future research includes extended follow-up periods (e.g., 6 or 12 months) to better assess the long-term outcomes of HBCR. These suggestions have been incorporated into the revised manuscript. The updated content can be found on page 13, paragraph 1, line 453-456. |
